# Bacterial load slopes represent biomarkers of tuberculosis therapy success, failure, and relapse

Gesham Magombedze [1✉], Jotam G. Pasipanodya[1,2] & Tawanda Gumbo [1,2]

There is an urgent need to discover biomarkers that are predictive of long-term TB treatment outcomes, since treatment is expense and prolonged to document relapse. We used mathematical modeling and machine learning to characterize a predictive biomarker for TB treatment outcomes. We computed bacterial kill rates, $\gamma_f$ for fast- and $\gamma_s$ for slow/non-replicating bacteria, using patient sputum data to determine treatment duration by computing time-to-extinction of all bacterial subpopulations. We then derived a $\gamma_s$-slope-based rule using first 8 weeks sputum data, that demonstrated a sensitivity of 92% and a specificity of 89% at predicting relapse-free cure for 2, 3, 4, and 6 months TB regimens. In comparison, current methods (two-month sputum culture conversion and the Extended-EBA) methods performed poorly, with sensitivities less than 34%. These biomarkers will accelerate evaluation of novel TB regimens, aid better clinical trial designs and will allow personalization of therapy duration in routine treatment programs.

---

[1] Center for Infectious Diseases Research and Experimental Therapeutics (CIDRET), Baylor Research Institute, Dallas, TX, USA. [2] Quantitative Preclinical and Clinical Sciences Department, Praedicare Inc, Dallas, TX, USA. ✉email: gmagombedze@gmail.com

Tuberculosis (TB) is the most important infectious cause of death worldwide, accounting for 3% of all deaths; it killed one billion people over the last two centuries[1]. In both drug-susceptible TB and multidrug-resistant TB (MDR-TB)[2], therapy duration is 6 months, after which patients are followed for up to 18 months to document relapse. The large numbers of patients with TB (10 million/year), the long therapy duration, and the follow up period of up to 2 years, makes TB one of the most expensive diseases to treat. Thus, it is of crucial importance to identify TB treatment regimens that are equally as effective in drug-resistant TB as in drug-susceptible TB, to identify regimens that can shorten therapy duration, and to identify early biomarkers that obviate the need for 2-year follow up[1–11]. A closely related problem is the time it takes to evaluate and compare such new regimens in phase I-III clinical trials; they take decades to complete given the long follow-up time required to document relapse. Thus, biomarkers that obviate the need for the long follow up to document relapse, and that can be deployed immediately on a global scale at little cost, need to be urgently developed for both routine patient care and to accelerate the time-table of clinical trials.

The tools currently used to monitor TB treatment in the clinic and in clinical trials arose in the historical context of the microbiology technology of 50 years ago. In the late 1970s Jindani and Mitchison performed a 14-day treatment clinical study in East Africa ($n = 124$ patients) that utilized solid agar-based *Mycobacterium tuberculosis* (*Mtb*) colony-forming unit (CFU)-derived kill rates defined by linear regression slopes to define early bactericidal activity (EBA), and the 14-day or extended-EBA to capture sterilizing activity, which are the basis of current phase II clinical trials[7,8]. In 1993 Mitchison summarized results of seven clinical studies to propose the use of two-months sputum culture and smear as a surrogate of relapse; the two-month (eight-week) endpoint is now the basis of clinical decision-making in routine clinical care[3,10–13]. Eight-week studies are also widely used as phase IIb studies to select TB regimens that go into the larger phase III studies in which long-term outcomes such as relapse, death, and cure are evaluated. However, the accuracy of these phase I/II studies in predicting hard clinical outcomes such as cure, therapy failure, and relapse, have been challenged[10–12,14,15]. In addition, more recent technological advances with semi-automated liquid cultures have demonstrated that the eight-week agar-based cultures may have been over-optimistic and are associated with substantial false-negative rates[16–19]. On the other hand, time-to-positivity (TTP) in the liquid cultures can be used in place of CFUs[20,21]. The liquid culture technology has been widely deployed across the world for routine clinical care as a diagnostic and for susceptibility testing. Here, we sought to identify mechanistic biomarkers (based on quantitative biology of the disease) that fulfill the definition of the US Food and Drug Administration BEST (Biomarkers, EndpointS, and other Tools) Resource, for use early during therapy to predict long-term hard clinical endpoints such as cure, therapy failure, and relapse[22,23].

We have developed a mechanistic model to quantitatively explain the drug-regimen bacterial kill kinetics and dynamics of both fast-replicating and semi-dormant/non-replicating persistent (NRP) *Mtb* subpopulations in TB patients as reflected in sputum[24]. Here, we used serial sputum TTP-data from patients in the Rapid Evaluation of Moxifloxacin in Tuberculosis (REMoxTB) phase III clinical trial to identify the trajectory of these two bacterial sub-populations and to estimate time in which both *Mtb* bacteria subpopulations reach extinction (time-to-extinction)[24]. According to Burman, "The ability to prevent relapse is termed sterilizing activity because it is presumed to require killing nearly of all bacilli remaining after the initial phase of therapy"[9]. Restated, failure to reach extinction by the *Mtb*

population in lung lesions is a required condition for therapy failure and relapse. Therefore, the time-to-extinction of all bacillary populations marks the required minimum duration of therapy in order to avoid relapse. However, some patients who do not reach bacterial-population extinction can still achieve relapse-free cure because of immune-response that can potential eliminant remaining bacteria.

## Results

**Clinical and laboratory characteristics in derivation and validation datasets.** First, REMoxTB clinical trial patients who had (i) majority of sputum samples that were contaminated (TTP<4days), or missing, or (ii) ≤4 data-points within the first 8 weeks (i.e., data-points fewer than ODE model parameters minus one) excluding the baseline value were removed, leaving 637 (33%) patients randomized to the standard therapy arm, 654 (34%) randomized to the isoniazid arm, and 633 (33%) randomized to the ethambutol arm (Fig. 1). This was followed by converting the 1,924 patients TTP-series to CFU/mL using Eq. 1, before modeling the data with a set of ODEs 2 and 3, to describe trajectories of *Mtb* CFU/mL with time (i.e., slopes). We identified ODE-model parameter estimates using 8-week (2-months)-, 4-months-, and 6-months accrued TTP-derived data for all 1,924 patients. The model parameter estimates are shown in Table S1. As an internal check for consistency with clinical observations, the range of proportions (fraction $f$) for semidormant and non-replicating bacteria to the log-phase population was 1% to 25%. We termed the *Mtb* kill rates $\gamma$-slopes, where $\gamma_f$ is the slope for fast-replicating *Mtb* and $\gamma_s$ is the slope for semi-dormant/non-replicating (NRP) *Mtb*. The model was also used to calculate the time-to-extinction of the total *Mtb* population for each patient, with results shown in Fig. 2.

**Data partitioning into derivation and validation datasets.** We separated the 1924 patients' data into derivation and validation datasets, shown in Table 1. The derivation dataset was comprised of 318 (50%) patients on standard therapy, as shown in Fig. 1. All patients in the derivation dataset were randomized to six-months therapy duration. The validation datasets comprised of (i) 319 patients on standard therapy for six-months duration, and (ii) 1287 patients randomized to the experimental arms (isoniazid or ethambutol) that had a four-months therapy duration. Table 1 shows that the demographic and clinical characteristics were similar between the derivation data set and all validation data sets, which means that the data-partitioning step was executed successfully.

**Time-to-extinction versus clinical trial-based outcome definitions.** We then used the derivation dataset to determine if the time-to-extinction of the total *Mtb* sputum population for each patient had clinical relevance, especially given that TTP versus CFU/mL relationship could change with time during treatment. The number of patients deemed cured at different time intervals in the course of treatment obtained by counting the number of negative cultures/TTP as defined in the REMoxTB protocol versus those identified using our time-to-extinction model definitions (derived from CFUs calculated from TTPs) had a Spearman rank correlation of 1.0 ($p = 0.017$). Moreover, when we used Cohen's kappa ($\kappa$) to assess agreement between individual pairs of either time-to-extinction versus standard clinical definitions, they were highly concordant ($\kappa = 0.65$, $p < 0.001$). Furthermore, the Spearman rank correlation between $\gamma_f$ (fast slope) and 14-day extended-EBA (derived using linear regression) was 0.68 ($p < 0.001$), which suggests that the extended-EBA mainly reflects

**Step 1**

> **REMoxTB TRIAL DATABASE**                                      N=1924 patients
>
> **Patient with sufficient serial sputum samples chosen**
> **-6 months duration standard therapy arm [N=637]**
> **-4 months duration isoniazid arm [N=654]**
> **-4 months duration ethambutol    [N=633]**

**Step 2**

> **REMoxTB TRIAL DATABASE**                                      N=1924 patients
>
> **Ordinary differential equations that track fast replicating and slow replicating**
> **bacteria subpopulations in lung lesions using the sputum TTP proxy**
>
> **-Output 1: Microbial kill slopes for fast replicating bacteria**
> **-Output 2: Microbail kill slopes for slow replicating bactetria**
> **-Output 3: Time-to-extinction of all bacteria subpopulations**

**Step 3**

> **REMoxTB TRIAL DATABASE**                                      N=1924 patients
>
> **Data partitioning into derivation dataset of 6-months standard**
> **therapy [N=318] and validation datasets [N=1606] (two 4-months**
> **experimental arms and the remaining half of the 6-months standard arm)**

**Step 4**

> **REMoxTB TRIAL DERIVATION DATABASE**                           N=318 patients
>
> **IDENTIFYING BIOMARKERS USING MACHINE LEARNING**
>
> **Stage 1**: Classification and regression trees [CART] to rank top
> predictors for time-to-extinction (TTE) defined outcomes
>
> **Stage 2**: Clustering of TTE defined outcomes versus top ranked
> CART predictors
>
> **Stage 3**: Monte-Carlo simulations identify biomarkers thresholds
> in indeterminate outcome zones
>
> **Stage 4**: Creation of a biomarker rule to predict outcomes for
> different therapy durations

**Step 5**

> **REMoxTB TRIAL PREDICTION/VALIDATION [6-months]**              N=218 patients
> **Sensitivity and specificity for the biomarker rule in patients on**
> **standard therapy using TB protocol definitions for clinical outcomes**

**Step 6**

> **REMoxTB TRIAL PREDICTION/VALIDATION [4-months]**              N=1063 patients
> **Sensitivity and specificity for the biomarker rule in patients in experimental**
> **therapy regimens using TB protocol definitions for clinical outcomes**
> **- Isoniazid arm [N=530]**
> **- Ethambutol arm [N=533]**

**Fig. 1 Biomarker development steps. Step 1**: Patients without sufficient data points to derive bacterial kill slopes were removed. **Step 2**: The weekly sputum time-to-positivity data was then converted to colony forming units and then modeled using ordinary differential equations. **Step 3**: Data partitioning of 50% of patients in standard of care six-months therapy as derivation data-set and the other 50% into valdiation dataset. All patients in experimental arm, administered over 4 months were assigned to validation datasets. **Step 4**: Four mathematical modeling and machine learning types of analyses in derivation dataset to[1] identify predictors of time-to-extinction (TTE) and[2] threshold values deliniating different TTE, and[3] design a diagnostic rule for different therapy durations. **Step 5**: Accuracy of diganostic rule/biomarker for six-months therapy duration in standard of care validation dataset using clinical definitions of outcome (relapse, cure). **Step 6**: Accuracy of diganostic rule/biomarker for four-months therapy duration in two experimental arms in validation dataset using clinical definitions of relapse and cure.

the effect of treatment on *Mtb* in logarithmic-growth phase and not semidormant/NRP bacilli, as was assumed in the past[8].

**Predictors of outcome in derivation dataset.** Classification and regression trees (CART) were used, to identify predictors of target outcome, defined as sputum microbial outcomes (cure at end of therapy, therapy failure or relapse), using potential predictors that included ALL the clinical and laboratory features, including ODE-model derived γ-slopes, for the tasks of classification and regression as input/independent variable. CART identified the $\gamma_s$ (semi-dormant/NRP kill) slope as the primary predictor (which had a variable importance score of 100%), followed by initial bacterial burden just prior to therapy commencement (which we termed $B(0)$), which had a variable importance score of 91.7%. This means that the initial TTP ($B(0)$) improved the primary predictor by an extra 91.7%. Notably, $\gamma_f$ was not ranked as a predictor using this agnostic machine learning method. Similarly,

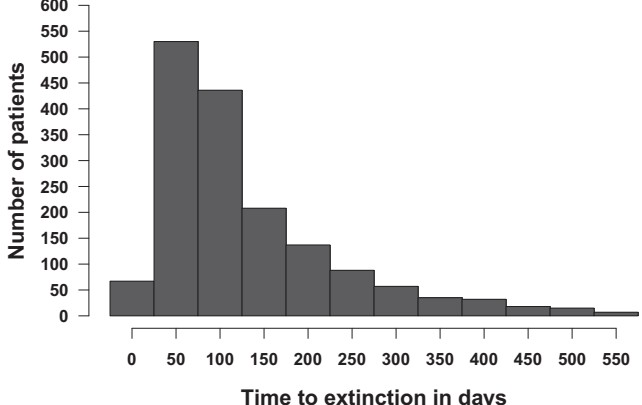

**Fig. 2 Distribution of time-to-extinction for all 1924 patients.** Shown is the data for all the 1646 patients who achieved bacillary population extinction; bacilli in the remainder of patients did not reach extinction, so the time is at infinity. The mean time to extinction and 95% confidence intervals were 122.4 (117.9 to 126.9) days.

features such as HIV status, cavitation, biological sex (male or female) ranked low and had variable importance scores below 10%. CART performs its own cross-validation within the derivation dataset, in this case by randomly splitting the derivation dataset five times. With the cross-validation, the post-test validation area under the curve (AUC) in the same derivation set was >85%, demonstrating that $\gamma_s$ plus initial TTP ($B(0)$) will likely perform as good predictors in future and separate datasets.

**Clustering-based approaches to identify biomarkers in derivation dataset.** Cluster analysis is an agnostic, quantitative and unsupervised machine-learning we used to group similar longitudinal patterns. In this case, TTP trajectories in the derivation dataset (with 238 patients) were grouped into four distinct homogenous groups based on the K-means algorithm as shown in Fig. 3. These were[1], a cure cluster of 80 (33.61%) patients (Fig. 3a, b)[2], a slow-cure cluster of 100 (42.02%) (Fig. 3c, d)[3], a relapse cluster of 34 (14.28%) patients (Fig. 3e, f), and[5] a treatment failure cluster of 24 (10.08%) patients (Fig. 3g, h). The slow cure cluster identified by this unsupervised machine learning method denoted those patients who had delayed attainment of microbiologic cure at the end of six months therapy (failed therapy at the end of six months) but achieved relapse-free cure when standard therapy was continued beyond six months duration. These four clusters represented 238/318 (74.84%) of patients with less than 2 missing observations or more missing observations during follow up. The model explained these data well, as is shown in online Figures S2 and S3, and Table S1, while the corresponding summary statistics for each cluster are shown in Table S2.

We used this clustering step to identify the minimum duration of data gathering that would give a γ-slope that could accurately predict cure or therapy failure or relapse. Figure 4 shows the distribution of model derived $\gamma_s$ and $\gamma_f$ values, when these slopes were derived based on 8-weeks-derived TTP data (2-months) (Fig. 4a, b), 4-months-derived TTP data (Fig. 4c, d), and 6-months-derived TTP data (Fig. 4e, f). The 8-week-, 4-months-, and 6-months-derived $\gamma_s$ and $\gamma_f$ values (shown in Table S1) versus outcomes were examined in pairwise comparisons using the Mann-Whitney-Wilcoxon test. Figure 4 shows that the $\gamma_f$ values did not

| Variable | Total sample | Derivation dataset | Validation datasets | | | *p*-value |
|---|---|---|---|---|---|---|
| | $n = 1924$ (%) | Standard therapy; $n = 318$ (%) | Standard therapy; $n = 319$ (%) | Ethambutol; $n = 633$ (%) | Isoniazid; $n = 654$ (%) | |
| Age, years [mean (SD)] | 33.40 (12.16) | 33.10 (11.93) | 33.81 (12.40) | 33.88 (12.15) | 32.89 (12.17) | 0.442 |
| **Sex** | | | | | | |
| Female | 585 (30) | 91 (29) | 101 (32) | 188 (30) | 205 (31) | 0.767 |
| Male | 1339 (70) | 227 (71) | 218 (68) | 445 (70) | 449 (69) | |
| **Race** | | | | | | 0.663 |
| Black | 861 (45) | 149 (47) | 146 (46) | 289 (46) | 277 (42) | |
| Asian | 586 (30) | 96 (30) | 96 (30) | 193 (30) | 201 (31) | |
| Mixed | 451 (23) | 66 (21) | 74 (23) | 142 (22) | 169 (26) | |
| Other | 26 (1) | 7 (2) | 3 (1) | 9 (1) | 7 (1) | |
| **Country site** | | | | | | 0.986 |
| China | 22 (1) | 6 (2) | 2 (1) | 5 (1) | 9 (1) | |
| India | 372 (19) | 58 (18) | 61 (19) | 126 (20) | 127 (19) | |
| Kenya | 136 (7) | 26 (8) | 18 (6) | 43 (7) | 49 (7) | |
| Mexico | 22 (1) | 7 (2) | 2 (1) | 8 (1) | 5 (1) | |
| Malaysia | 69 (4) | 10 (3) | 13 (4) | 20 (3) | 26 (4) | |
| Thailand | 119 (6) | 21 (7) | 19 (6) | 41 (6) | 38 (6) | |
| Tanzania | 211 (11) | 37 (12) | 37 (12) | 73 (12) | 64 (10) | |
| South Africa | 908 (47) | 142 (45) | 156 (49) | 297 (47) | 313 (48) | |
| Zambia | 65 (3) | 11 (3) | 11 (3) | 20 (3) | 23 (4) | |
| Sputum TTP in days (SD) at start of therapy | 5.16 (1.21) | 5.26 (1.16) | 5.13 (1.26) | 5.15 (1.20) | 5.13 (1.21) | 0.420 |

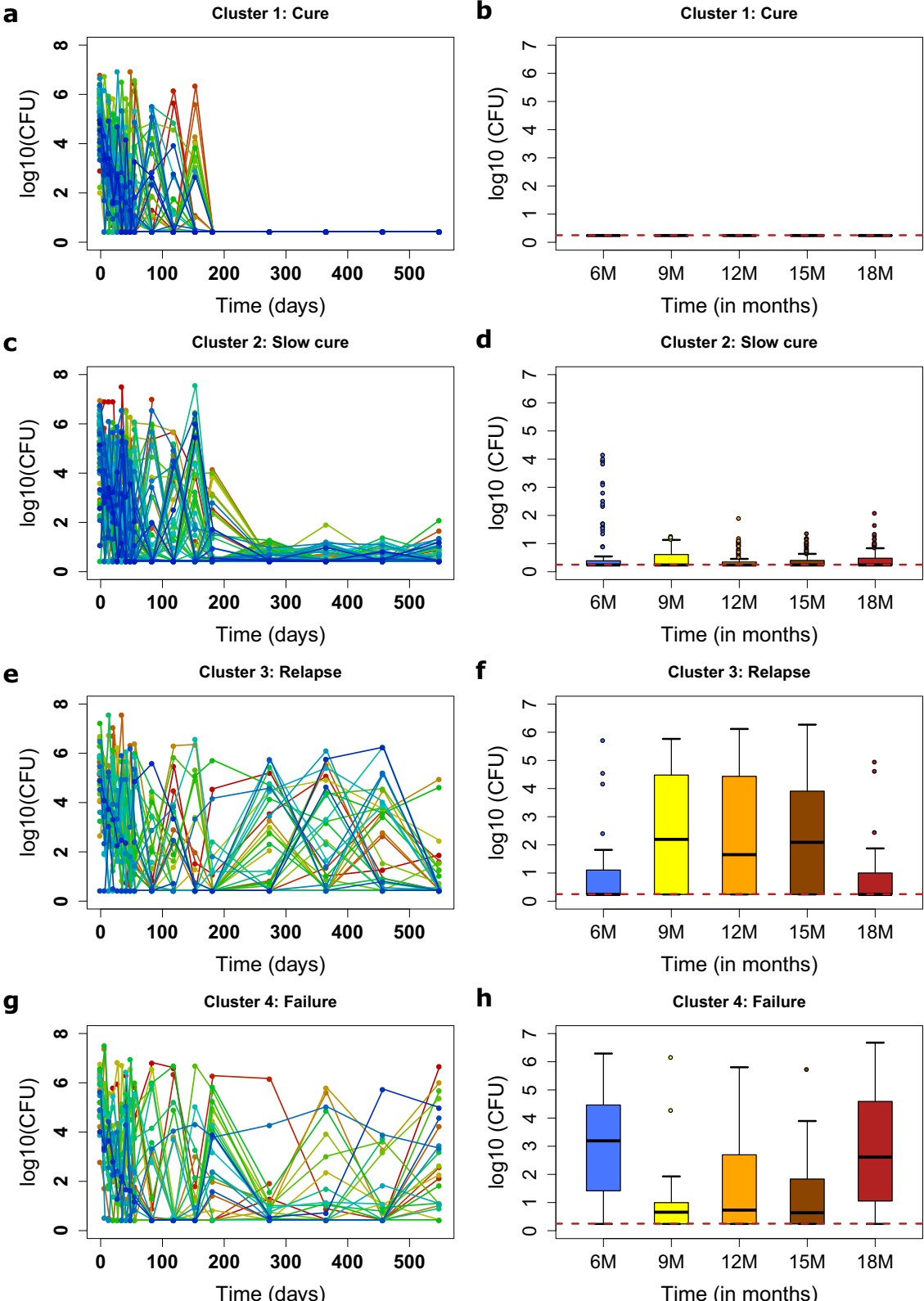

**Fig. 3 Clusters of treatment outcomes.** Clusters of CFU/TTP trajectories of each individual patient are shown side by side with median $\log_{10}$ CFU/mL plus interquartile range, for the follow up periods of 6 to 18, months. **a** Cured patients' trajectories and **b** summary of trajectories during follow-up. **c** Slow cure trajectories and **d** box-plots of CFUs after therapy completion. Relapse patterns **e** and the corresponding patterns during follow up **f**. Failed treatment cluster **g** and follow-up period summarizing boxplots **h**.

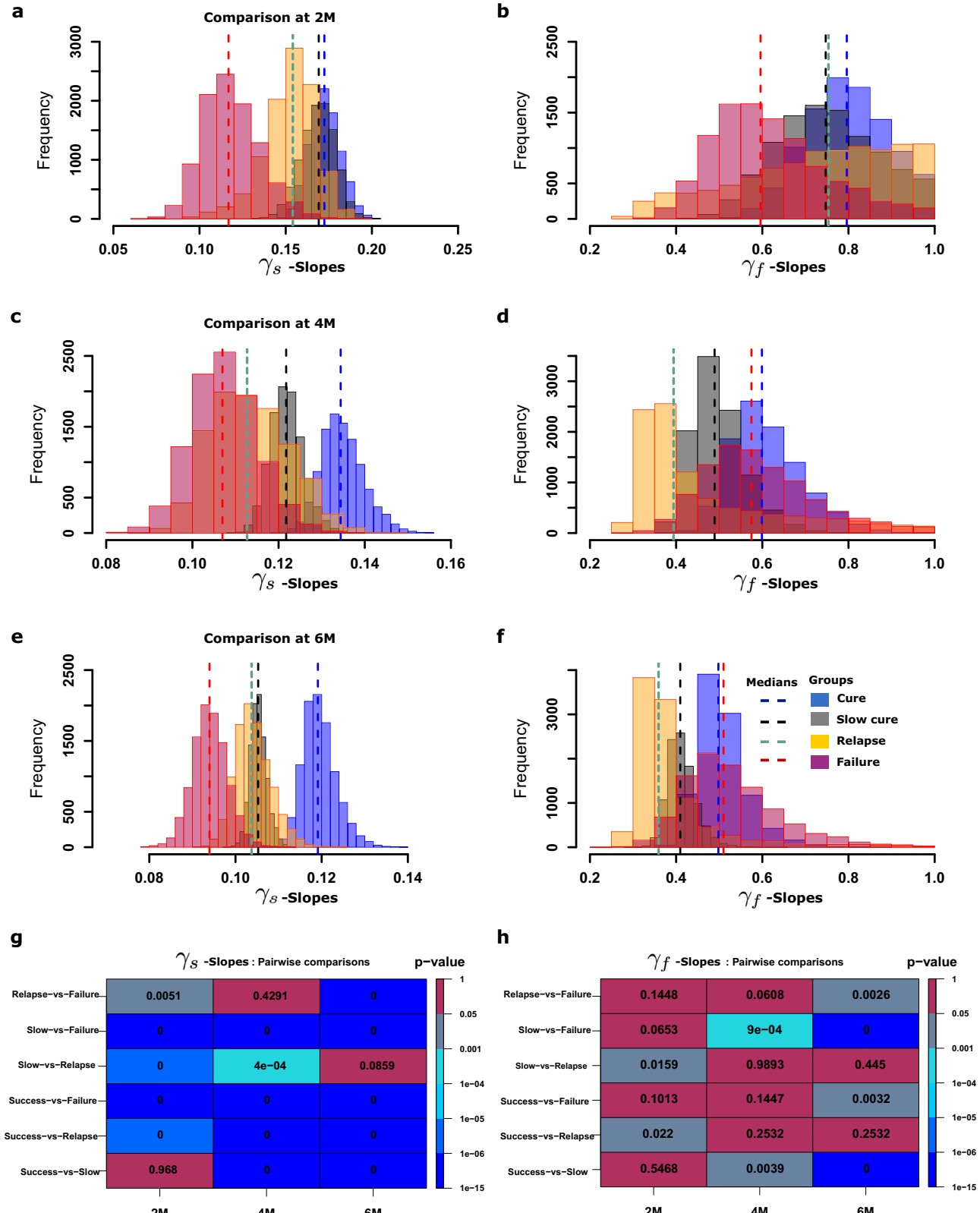

**Fig. 4 Distributions of model estimated kill rates/biomarkers.** Distributions of $\gamma_s$ - and $\gamma_f$ -slopes based on 2-months, 4-months and 6-months accrued TTP data. **a** $\gamma_s$ -slopes based on 2 months TTP data, **b** $\gamma_f$ -slopes based on 4 months TTP data, and **c** $\gamma_s$ -slopes based on 6 months TTP data. **d** The magnitude of $\gamma_f$ -slopes at 2 months, **e** at 4 months, and **f** at the end of 6 months. **g** Pairwise analysis using the Mann–Whitney test for distributions of slopes based on 2, 4 and 6 months data versus each cluster (cure, slow-cure, relapse and failure) for the $\gamma_s$ -slopes shows $p$-values in each cell that were significant even with 2-months of data, except with cure versus slow cure group. **h** Pairwise statistical difference analysis for $\gamma_f$, found that few $p$-values were significant, and even those had an inconsistent pattern.

discriminate failures from cures, consistent with CART findings. However, $\gamma_s = 0.15$ or $<0.1$ log$_{10}$ CFU/mL/day (modeling semi-dormant/NRP *Mtb*) were better at discriminating these outcomes. The slopes derived with 8-week-vs-4 months data differed in the misclassification of patients' outcomes, the former misclassifying more relapses as cures and the latter misclassifying more cures as relapses. Nevertheless, as demonstrated by the statistical comparisons in Fig. 4h, the 8-week derived TTP data $\gamma_s$ (Fig. 4g) adequately diagnosed relapse versus other outcomes. In other words, $\gamma_s$ calculated using eight-week-derived TTP data is a good predictor of sterilizing effect up to 18-months after therapy cessation, and this eight-week data-derived slope thus measures sterilizing activity rate. Subsequently, all $\gamma_s$ discussed herein were those identified using the first eight-weeks-derived data.

**Monte Carlo Simulations to identify biomarker thresholds in indeterminate outcome zones.** Given the misclassification of relapses as cures by the eight-week TTP-derived $\gamma_s$, and $\gamma_s$ thresholds in the indeterminate outcomes region (i.e, overlap of relapse versus slow cures), we utilized Monte Carlo Simulations (MCS) of time-to-extinction in tandem with CART to further discriminate $\gamma_s$ cut-off values in indeterminate outcome zones, with results shown in Table S3, Figures S4 and S5. Cure was clearly delineated by $\gamma_s > 0.15$, therapy failure by $\gamma_s < 0.1$ plus initial bacterial burden $B(0) > 5.6$ log$_{10}$ CFU/mL (TTP = 5.49 days), and relapse delineated from cure by $\gamma_s < 0.13$. Figure S4c, d shows the CART-derived biomarker thresholds based on the simulation for predicting treatment outcomes after either 4-months or 6-months therapy duration. Patients with initial bacteria burden $B(0) > 4.5$ log$_{10}$ CFU/mL (TTP = 8.11 days), and $\gamma_s$-slopes between 0.1 and 0.15 had >55% chance of failing treatment at 6 months (Fig. S4c). However, for a four-month therapy duration regimen, patients with $B(0) > 5.4$ log$_{10}$ CFU/mL (TTP = 5.93) and $\gamma_s$ between 0.09 and 0.14 had a > 65% chance of failing treatment. Figure S4 also shows that in order to achieve cure/bacillary population extinction within 2 months of treatment, then $\gamma_s \geq 0.15$ ($-3.90$ TTP per day) would be required, while patients with $\gamma_s \leq 0.1$ ($-2.60$ TTP per day) would fail. Patients on standard therapy with $B(0) > 5.6$ log$_{10}$ CFU/mL (TTP = 5.49) with $\gamma_s < 0.13$ would relapse.

**Creation of $\gamma_s$-based rule to predict relapse for different therapy durations from derivation dataset.** In the final derivation step, we established a diagnostic rule for the relationship between $\gamma_s$ -slopes and the outcomes, using Latin hypercube sampling for sensitivity analyses, with results shown in Fig. 5. Figure 5a–d shows that increasing or reducing the $\gamma_s$ (i.e., speed of kill of slow-replicating bacteria) $\gamma_s$ changes the time-to-extinction and therefore the required minimum duration of therapy. As an example, the six-months therapy duration would need to be extended to eight-months duration (i.e., slow-cure) in patients with high bacterial burden when $\gamma_s$ is reduced from 0.148 to 0.131 and extended to 9-months when its reduced to 0.125 (Fig. 5b). However, for patients in the medium and low CFU load categories, lower slopes can still achieve cure within 6 months (Fig. 5c, d). On the other hand, to reduce treatment duration to four-months $\gamma_s$ should increase to 0.183, and in order to reduce therapy duration to two-months $\gamma_s$ should increase to 0.286 (Fig. 5b). The relationship between $\gamma_s$ and initial TTP versus minimum duration of therapy is shown in Fig. 5e, f, and is non-linear function. From this, we calculated the target $\gamma_s$ to achieve cure (extinction of bacterial population) with one-month therapy duration, shown in Fig. 5e, f. This establishes a diagnostic rule between $\gamma_s$ versus minimum treatment duration for relapse-free cure for different initial *Mtb* burdens. After this step, the derivation work was completed, and the derivation dataset patients excluded from subsequent validation studies.

**Performance of $\gamma_s$-based rules in forecasting outcomes for 6 months therapy duration.** Next, we calculated the accuracy of how well our diagnostic rule performed in the six months therapy duration validation datasets, using the clinical and microbial treatment outcomes defined by the REMoxTB trial protocol. Treatment outcome calls could be made in 218 of the 319 patients who also had more than 4 data points within eight-weeks to give statistically robust estimates of the bacteria kill slopes: 169/218 (74.31%) achieved relapse-free cure, 137/218 (16.97%) had therapy failure at the end of treatment, and 19/218 (8.72%) relapsed after initially looking like cure at the end of therapy. The accuracy of the $\gamma_s$-based rules are compared to the extended-EBA and two months sputum conversion in Table 2, together with the relative risk (RR) of failure when each biomarker was positive versus not-positive (numbers in each cell shown in Table S4). Table 2 shows that the extended-EBA had a sensitivity of 14% and specificity of 92% in identifying failure from cure without relapse and the RR 95% confidence interval crossed 1 ($p = 0.205$); the number needed to diagnose (NND) failure/relapse was 15.27. Similarly, two-months sputum conversion had a sensitivity of 33% and specificity of 71%, RR was statistically 1, and NND was 21.41. On the other hand, the eight-weeks-data derived $\gamma_s$ combined with the initial TTP at treatment commencement had a sensitivity of 92% and specificity of 86% in identifying failure from relapse-free cure, the RR of failure when this biomarker was positive versus not positive was 28 (Table 2 and Table S4), while NND was 1.29. Failures either arise as therapy failure or relapse; Table 2 shows the sensitivities for these different biomarkers in predicting relapses from treatment failures. The slope decision rule based on $\gamma_s > 0.15$ has a sensitivity of 92% and a specificity of 89% in predicting relapses from failures. Thus, the biomarkers we derived were highly specific at identifying relapse-free cure, therapy failure, and relapse.

**Performance of $\gamma_s$-based rules in forecasting 4-months therapy duration outcomes.** Also, we tested the accuracy of the diagnostic rule for four-months therapy duration in the validation datasets comprised of the REMoxTB trial experimental arm patients. In the arm in which isoniazid was replaced by moxifloxacin and therapy administered for four-months ($n$=655), 530 patients had enough TTP data in the first 8 weeks to calculate slopes. In this dataset, 369/530 (69.62%) patients achieved cure, 40 (7.55%) patients had therapy failure at the end of 4 months of therapy, and 121 patients (22.83%) relapsed. Table 2 shows that the $\gamma_s > 0.15$ had a sensitivity of 81% and specificity of 89% for relapse-free cure versus failure, and among the failures had a sensitivity of 75% and specificity of 60% for separating relapse from therapy failure. The relative risk of failure in patients with positive slope-based biomarker versus negative biomarker was approximately 15 (Table 2 and Table S4); the NND was 1.47. The 2-month sputum conversion was not designed for 4 months therapy duration regimens, and is not shown, while the extended-EBA which is used to triage shorter duration regimens is shown; the NND was 16.69.

In the arm in which ethambutol was replaced by moxifloxacin ($n$=633), 533 patients had enough data to calculate 8-week slopes. In this dataset 385 (72.23%) of patients achieved cure, 46 (8.63%) had therapy failure, while 102 relapse (19.4%). The sensitivity of the extended EBA was only 10%, and the NND was 18.73. The sensitivity of $\gamma_s$-based slopes was 70% and the specificity 71% for cure versus therapy failure, while the sensitivity was 70% and specificity 65% for picking relapse versus therapy failure. The NND was 1.89.

In order to summate, we calculated an overall value of the relative risk of failure when our $B(0)$ and $\gamma_s$-based slope predicted poor outcome for a specified duration of therapy (using 6-months and 4-month duration data combined). Among patients with positive

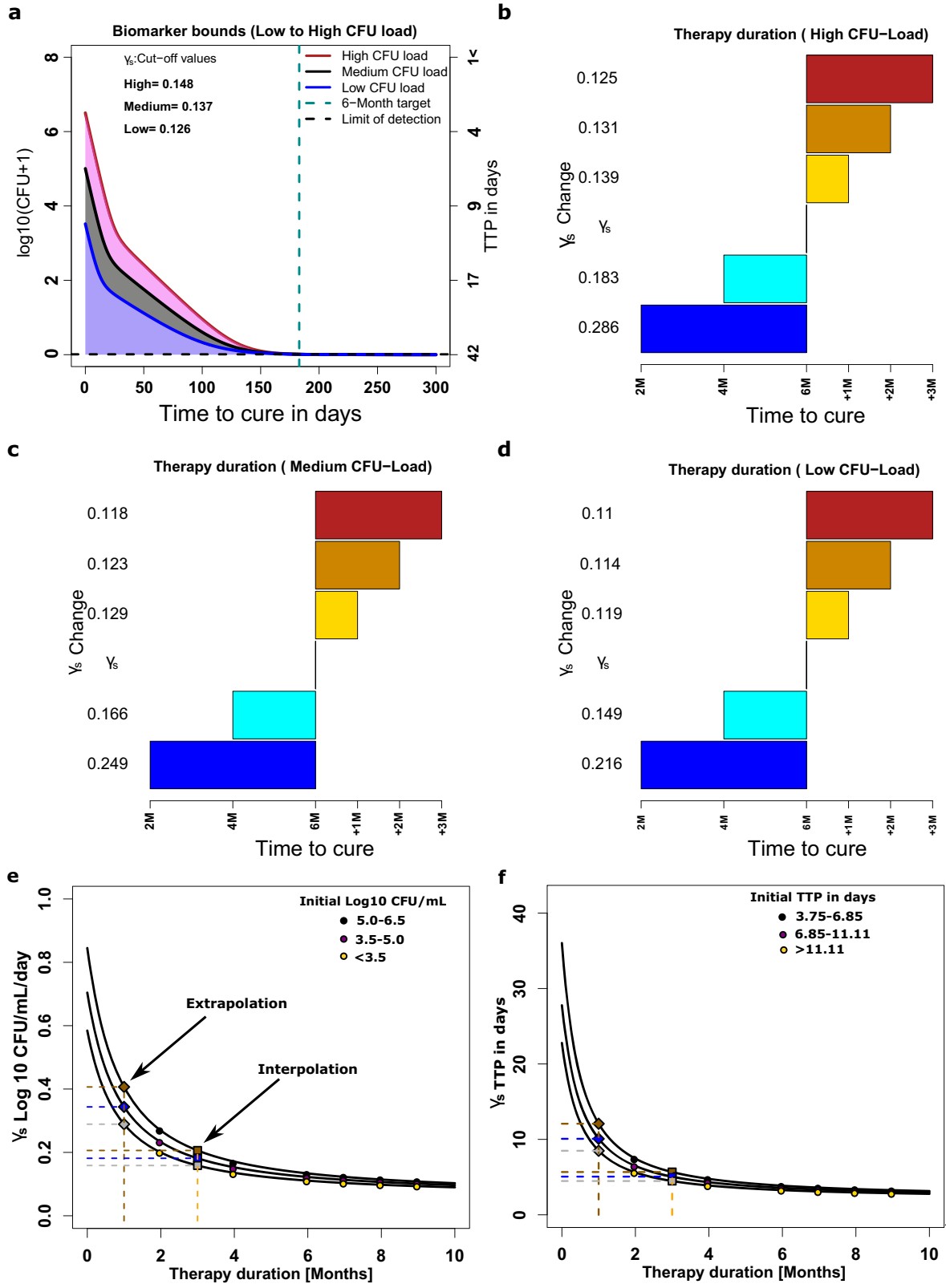

biomarker for specified therapy duration, 159/205 (78%) failed therapy compared to 218/1072 (20%) in whom the biomarker was negative. The RR of failure with the rule was 8.25 (95% CI: 6.09–11.20); *p*<0.0001. In terms of cure only 4% of entire validation dataset cohort of patients achieved relapse-free cure when our rule was positive while 67% achieved cure when it was negative.

## Discussion

First, our model estimates of initial proportion of non-replicating persistent/semidormant of 1-25% of fast replicating (Table S1), and the microbial kill slope inflection point of about 14 days would seem to be different from the proportions of ~0.1% NRP (slow) of total bacilli and early bactericidal effect studies in which the

**Fig. 5 Sensitivity analyses and rule-making of $\gamma_s$ slopes versus time-to-cure. a** Shown are $\gamma_s$ slopes required to achieve cure within 6 months for patients with high bacterial burden compared to those with medium and lower bacterial burdens. **b–d** The $\gamma_s$ slopes required to achieve cure at 2, 4 and 6 months duration or for delayed cure of an additional 1 to 3 months beyond month 6 (i.e, 6 months +1, or +2, or +3 months), are shown for patient starting with high (**b**), medium (**c**), and low (**d**) *Mtb* burdens. **e** Magnitudes. of slopes for therapy duration of only 1 and 3 months (for high, medium and low *Mtb* burden) could be extrapolated and interpolated in log $_{10}$ CFU/mL/day as (0.42, 0.36, and 0.30) and (0.22, 0.20, and 0.17), respectively, based on the relationships between slope and duration of therapy ($r^2 > 0.999$). **f** Magnitudes of slopes for therapy duration of 1 and 3 months are extrapolated and interpolated TTP-slope as (12.08, 10.08 and 8.49) and (5.67, 5.06 and 4.48), respectively, based on the relationships between TTP-derived slope and duration of therapy ($r^2 > 0.999$).

---

### Table 2 Biomarker threshold values, sensitivity and specificity scores, and risk of failure, with 95% confidence intervals.

| Biomarker using log$_{10}$ CFU/mL | Sensitivity | Specificity | Accuracy | Relative risk of failure with positive biomarker |
|---|---|---|---|---|
| **Six-months therapy duration** | | | | |
| Extended EBA: Failure vs Cure | 0.14 (0.05–0.30) | 0.92 (0.87–0.96) | 0.79 (0.73–0.84) | 1.71 (0.73–3.48) |
| 2 months smears/culture: Failure vs Cure | 0.33 (0.17–0.53) | 0.71 (0.64–0.78) | 0.65 (0.58–0.72) | 1.20 (0.60–2.34) |
| $\gamma_s^1$ slope (<0.1): Failure vs Cure | 0.57 (0.39–0.73) | 0.95 (0.91–0.98) | 0.89 (0.84–0.93) | |
| $\gamma_s^2$ slope (>0.15): Failure vs Cure | 0.91 (0.78–0.98) | 0.86 (0.79–0.90) | 0.87 (0.81–0.91) | |
| Slow slope plus $B(0)$: Failure vs Cure $\gamma_s^1, \gamma_s^2, B(0)$ | 0.76 (0.59–0.88) | 0.89 (0.83–0.93) | 0.87 (0.81–0.91) | 29.84 (10.20–89.07) |
| Slow slope plus $B(0)$: Failure vs Relapse $\gamma_s^1, \gamma_s^2, B(0)$ | 0.92 (0.78–0.98) | 0.89 (0.67–0.99) | 0.91 (0.80–0.97) | 20.40 (7.17–58.08) |
| **Four months therapy duration** | | | | |
| ***Isoniazid arm*** | | | | |
| Extended EBA: Failure vs Cure | 0.10 (0.04–0.21) | 0.95 (0.93–0.98) | 0.84 (0.80,0.87) | 2.14 (0.99–3.99) |
| $\gamma_s^1, \gamma_s^2, B(0)$: Failure vs Cure | 0.81 (0.70–0.90) | 0.87 (0.83–0.90) | 0.86 (0.83–0.89) | 14.51 (8.33–25.41) |
| $\gamma_s^1, \gamma_s^2, B(0)$: Failure vs Relapse | 0.75 (0.58–0.87) | 0.60 (0.51–0.69) | 0.64 (0.56–0.71) | 3.15 (1.65–6.01) |
| ***Ethambutol arm*** | | | | |
| Extended EBA: Failure vs Cure | 0.10 (0.05–0.19) | 0.94 (0.92–0.96) | 0.79 (0.76–0.83) | 1.66 (0.89–2.81) |
| $\gamma_s^1, \gamma_s^2, B(0)$: Failure vs Cure | 0.70 (0.60–0.79) | 0.71 (0.67–0.75) | 0.71 (0.67–0.75) | 4.10 (2.78–6.08) |
| $\gamma_s^1, \gamma_s^2, B(0)$: Failure vs Relapse | 0.70 (0.59–0.79) | 0.65 (0.55–0.74) | 0.68 (0.60–0.74) | 2.07 (1.50–2.87) |

TTP (0) is the corresponding TTP in days for $B$ (0): A dash means no cut-off value evaluated. The thresholds for predicting relapses-vs-cure are multiple steps however, are with the 0.1 to 0.15 indeterminate regions of the slow slope cut-offs for screening cures and failures.

---

inflection point is 3 days[7–9,25–27]. However, in studies that have directly visualized sputa of patients using lipid bodies (LB) to identify slow bacilli (and thus nicknamed "fat and lazy" by Garton et al.) and confirmed by bacterial *tgs1* gene expression, "the frequency of LB–positive cells varied from 3% to 86%"[28]. Estimates of LBs by Sloan et al were 28% (interquartile range: 13–54)%[29]. Thus, our model estimates of proportion of slow bacilli are in the same range as LBs visualized in sputum. The Sloan study also followed the change in LBs on standard therapy and identified the inflection point in proportion LBs was after 14-21 days, consistent with our inflection points slow bacilli in Figure S2. Thus, our model parameter estimates were highly consistent with the biology of bacillary populations, and their changes with therapy. This is the major strength of our approach, which is that they are mechanistic and based on the biology observed in patients.

Second, we found that the $\gamma_s$ (slow replicating) slope is a good surrogate of sterilizing activity, based on ability to predict relapse. Conversely, the extended EBA had a sensitivity of 14% for predicting outcomes at 6 months and beyond, and a poor accuracy. The extended EBA is effectively two-weeks accrued data; the poor sensitivity means that the total time for which the bacterial kill data is collected is too short to accurately capture sterilizing activity slopes. Indeed, the poor sensitivity of $\gamma_f$–slope-based metric means that most regimens with good sterilizing effect could be thrown away (too many false negatives for sterilizing activity) in regimen selection for sterilizing activity. Similarly, the 2-month sputum conversion had a sensitivity of 33% and specificity of 71%. These commonly used clinical indices gave us an

opportunity to externally validate our modeling approach. In this case, the last major meta-analyses on 2-month cultures as a predictor of long-term outcome in TB performed by Horne et al in 2010 identified a sensitivity of 40% (95% CI, 25–56%) and specificity of 85% (95% CI, 77%–91%), which was confirmed in subsequent studies[14,30,31]. Thus, our modeling findings are consistent with results of these major meta-analyses. This means that our 8-weeks-derived $\gamma_s$ slope plus initial bacterial burden, which had a sensitivity of 92% and specificity of 86% for 6 months therapy duration regimens, would perform better than the 2-month sputum conversion. In addition, our $\gamma_s$ slope can predict outcomes at shorter therapy durations than 6 months such as 4-months duration; the relative risk of therapy failure among patients with positive biomarker for specified therapy duration was > 8.0. Thus the $\gamma_s$-slope based on the first 8-weeks TTP data is a good response biomarker for sterilizing activity, even for therapy duration less than standard short course chemotherapy.

The $\gamma_s$-slope, which we will henceforth term the "sterilizing activity rate", fulfills the BEST criteria and definition of a monitoring biomarker in the category of a pharmacodynamic/response biomarker, in a similar fashion to HIV and hepatitis C viral load biomarker, and could play the same role in TB therapeutics and clinical trials[23,29,32,33]. According to BEST criteria, a pharmacodynamic/response biomarker provides early evidence (in this case 8-weeks) that a treatment might have an effect on a later pharmacologic clinical endpoint (in this case relapse at 2 years). In the case of HIV treatment trials, identification of viral load as a surrogate of efficacy in 1995 dramatically cut the duration and costs of clinical

trials, while avoiding use of potentially catastrophic clinical end-points such as therapy failure and death[32]. For TB, we propose identification and ranking of regimens using preclinical models that can accurately translate the sterilizing activity rate to patients[24,34]. The regimens so derived, including optimal doses, and the translated sterilizing activity rate will provide good priors for the design of 8-week clinical trials for novel regimens versus standard therapy, with weekly TTP as the main output and drug pharmacokinetics as a secondary outcome. The sterilizing effect rate ($\gamma_s$-slope), initial TTP, and trajectories can then be used to estimate therapy duration for the novel regimens and determine if indeed the new regimens can shorten TB treatment prior to performance of phase III studies. The 8-weeks TTP-data derived slopes can be used to compute a lower and more accurate patient sample sizes required to power the phase III trials, given the good accuracy in forecasting relapse. As an example, the number needed to diagnose (NND) failure and relapse of <2, when compared to ~20 for extended EBA and 5-6 for 2-months therapy, gives a more straightforward insight into the relative number of patients tested in each arm by different biomarkers. Moreover, these data and slopes can be used for optimal design by identifying optimal sampling times of sputa (TTP), and optimal number of samples that minimize uncertainty in the slopes in future clinical trials. Furthermore, since the predictive value of the sterilizing activity rates on relapse or cure or therapy failure is independent of the regimen the slopes can be used in clinical trials of MDR-TB and for "pan-susceptible" TB regimens, indeed for any TB regimen.

As regards to clinical practice, our findings add to the recent discovery that initial *Mtb* burden can be used to determine patients who can benefit from 4-month duration therapy[35]. Imperial et al used standard inferential statistics found that non-adherence was the most significant factor leading to poor outcomes (hazard ratio 5.9), and that low initial bacterial and disease burden were the most important determinants of optimum duration of therapy[35]. Here, we found that the sterilizing activity rate was ranked higher than initial bacterial burden or any other clinical factors. To put this is context, the risk of development of AIDS and death in patients whose HIV viral load did not reach undetectable within first 12 months was 2.40-fold compared to those who had, and a <75% reduction in viral load had a RR of 2.27-fold for poor outcomes[36–38]. Patients in whom the $\gamma_s$-slope-based rule was positive for different durations of therapy had a an 8.25-fold higher risk of failure, which is better performance than this commonly used HIV test used to individualize therapy. Thus, our findings could also be used to individualize therapy, in place of two-month smears/cultures currently recommended in routine care in TB programs worldwide. First, if these patients with potentially higher rates of therapy failure and relapse were identified during the first eight weeks of therapy, then interventions such as dose increases or switching therapy regimens could be made[39]. Second, the sterilizing effect rate ($\gamma_s$ slopes) could also be used by TB programs to identify patients who could be cured with specific shorter therapy durations of either 2, 3 or 4 months, on any regimen. Alternatively, they could be used to identify how long therapy duration should be extended beyond 6 months, thereby individualizing therapy duration, in patients with sputum $\gamma_s$ slopes that predict the slow cure clusters. Since many TB programs across the world already employ liquid culture systems that generate TTP, it means that the biomarker we propose would come at no extra cost to those TB programs. Computation of the slope could easily be implemented on a computer (or on a phone with specifically designed app).

Our study has some limitations. First, there were no accompanying CFUs to the TTPs in the REMoxTB study and we relied on the TTP to CFU conversion formula from a prior study;[24] REMoxTB TTP-CFU data-pairs would have been the best at characterizing the uncertainty in TTP conversion to CFU conversion. However, we tested the time-to-extinction based definitions of cure derived using this TTP-CFU conversion from the prior study to those observed using the REMoxTB clinical trial protocol definitions and patient microbiology and identified a Spearman rank correlation of 1.0 ($p = 0.017$). Thus, our TTP-CFU conversion and slopes derived from it were robust. Second, it could be argued that our findings are specific to the dataset we analyzed. However, the machine-learning cross-validation procedures we used are scored on how well predictors will perform on an entirely independent dataset in the future. Nevertheless, the accuracy of the biomarkers will still need to be further confirmed in other large datasets in a range of clinical contexts and with different regimens. Third, calculation of slopes is relatively complex. However, software can easily be written to automate this, as we have attempted elsewhere. Fourth, we relied on data supplied by the clinical trial team, and thus could not assess the quality and volume of sputum samples as treatment progressed, and several factors associated sputum collection effect of bacterial burden. This limitation however is somewhat mitigated by the finding that model fits did not change from start of therapy to end of therapy. Finally, not all patients who do not reach bacterial population extinction will fail therapy or relapse. This means that our approach may lead to over treating of these patients who would otherwise be cured. Examination of our proposed biomarkers with other tests such as radiological findings and therapeutic drug monitoring could reduce the number of over treated patients and are subject to ongoing analyses. However, even with these limitations, the early TTP-based biomarkers that we identified as predicting long-term clinical outcomes such as relapse for different therapy durations, have sensitivities and specificities that are higher than currently employed methods.

## Methods

**Study design, data extraction and definitions**. Our study design is reported in detail in Fig. 1. Briefly, we took data for bacteriologically confirmed TB patients that were enrolled in the REMoxTB clinical study[3]. In which patient sputum was cultured in the Mycobacteria Growth Indicator Tube (MGIT) to confirm bacteria viability. Since our aim was to develop a method agnostic of regimens used and drug-resistance status, patient data from the study[3] was used in our analyses regardless of drug-resistance status. Patients with majority of sputum samples that were contaminated or missing were excluded.

Patient and microbial details, including therapy regimens and serial TTPs, were extracted from the CPTR website (http://www.cptrinitiative.org). Time-to-extinction was defined as achieving a bacterial burden ≤$10^{-2}$ colonies/mL, as mathematically justified in our prior work[24]. Microbiologic cure was defined as two negative sputum cultures without an intervening positive. Relapse was defined by the re-appearance of positive culture in patients deemed cured at the end of therapy. Relapses were confirmed by 24-locus mycobacterial-interspersed-repetitive-unit analysis[3]. Failure to attain microbiologic cure at the end of therapy defined therapy failure, as per REMoxTB study protocol[3].

**Data partitioning**. Patients on the standard TB therapy regimen were randomly partitioned into two subsets of equal size. The first set was designated as the model derivation set, while the remainder was assigned for use in model validation (validation data set). To capture sufficient relapse events, only patients with at least two consecutive sputum samples during follow-up after treatment were used in model training and cross validation. Patients who received the experimental REMoxTB arms were used only in the validation dataset for sensitivity and specificity of predictors with 4 months therapy duration.

**Mathematical modeling for converting TTPs to CFUs**. In order to convert TTPs to CFU/mL, we applied the formula:

$$F = \alpha e^{-\beta x + \gamma} \qquad (1)$$

where $F$ is CFU/ml, $x$ is TTP (number of days the MGIT indicate presence of viable bacteria), α-represent $\log_{10}$ CFU/mL quantity when TTP take less than 1 day to turn positive, while $\beta$ is a rate (per day) that apportions changes in TTP to the corresponding CFU value, and $\gamma$ is a conversion adjustment parameter. Estimated values for these parameters are given in Table S5 together with an alternative model with $\gamma = 0$. We previously derived these parameters using more than 600 data point pairs from logarithmic phase growth and semi-dormant (or non-replicating phase)

hollow fiber system model experiments[24]. Bacterial burden from these experiments were quantified using (i) solid agar culture for CFUs, and (ii) liquid medium in the MIGIT for TTP. The hollow fiber model is repetitively sampled for CFUs and TTPs for up to 56 days on therapy. Bowness and colleagues have found that as treatment progresses, the recovered Mtb grew more slowly in culture, so that a linear equation model (including only constants a, b, c) that remain unchanged during treatment would be incorrect by day 14, and instead a Gompertz model with a time parameter would be better[40]. While our formula is not a linear regression equation, we still wanted to find out if it was accurate at the start of therapy as at 56 days, in patients. Therefore, we applied formula/equation #1 to an independent clinical data set of patients on TB therapy, the vitamin A study in which we had weekly TTPs and CFUs in 56 patients on standard therapy[18,24]. Results are shown in Figure S1, which shows that our formula remained accurate at 56 days as on day 0. Therefore, we employed equation #1 for toggling between CFU/mL and TTP.

**Mathematical model**. Our mathematical model, described in detail in the past[24], recapitulates events (i.e, Mtb burden) at site of infection, and, assumes two bacterial phenotypic populations: $B_f$, fast replicating bacteria in log phase, which grows at rate $r_f$ and $B_s$, non-replicating persisters which bacteria grow at rate $r_s$, such that where $r_f > r_s$, as observed by Canetti, McDermott et al, Sloan et al, Eum et al, and formalized by Mitchison[25–28,41,42]. Our assumption is that, in the lungs or at the site of infection, Mtb populations exhibit different physiological states, but share the same maximal bacterial burden, $K_{max}$[43,44]. The parameters $r_f$ and $r_s$ also measure of the reproductive or growth fitness, a measure of their virulence. The fast replication (log phase growth) Mtb grow at rate $r_f$ while the slow at rate $r_s$. It has been shown that in TB patients, these bacteria subpopulations co-exist, however, in active TB disease, the population of bacteria in log-phase is dominant[25,26,28,41,42,45].

$$\frac{dB_f}{dt} = r_f B_f \left( 1 - \frac{B_s + B_f}{K_{max}} \right) - \gamma_f B_f, \tag{2}$$

$$\frac{dB_s}{dt} = r_s B_s \left( 1 - \frac{B_s + B_f}{K_{max}} \right) - \gamma_s B_s. \tag{3}$$

The model has flexibility to track the time evolution of both Mtb subpopulations simultaneously, under effect of treatment with different combination regimens. In relation to assessing new surrogate markers or biomarkers for predicting TB treatment outcomes, the model has two sets of quantifiable parameters (i) $r_f$, $r_s$ and $K_{max}$ (Mtb growth parameters) and (ii) $\gamma_s$ and $\gamma_f$ (drug-regimen based microbial kill slopes), that are linked to disease pathogenesis, and therefore has the ability to predict disease outcomes independent of a specific TB therapy regimen. Further mathematical details and assumptions of the model are shown in our previous study[24].

**ODE-based model to data fitting**. First, all patient TTP longitudinal observations were converted to CFU values using Eq. 1. Then the data was fit to the system of ODEs (Eqs. 2 and 3). We implemented the Markov chain Monte Carlo (MCMC) method in R[22,24,46] to estimate the drug kill parameters using 50,000 runs of the chain. A Gaussian log-likelihood was used to generate posterior distributions for parameters assuming uniform distribution for the priors. Model to data fitting was done in two steps as was done in the study in Magombedze et al.[24]. In the first step, Hollow fiber control experiments data was used to estimate growth rates ($r_f$ and $r_s$) of the bacteria subpopulations and the bacteria population carrying capacity ($K_{max}$). The control experiments were carried out to determine the growth of the clinical isolates to imitate bacteria log-phase growth and the non-replicating growth phase. The model was fit to the data with the assumption that there were no drugs, that is, $\gamma_f = 0$, and $\gamma_s = 0$. The estimated values are given in Table S6. In the second steps, bacteria growth rates and carrying capacity estimated is step I, were used here and are kept fixed during model fitting to estimate the $\gamma_f$ and $\gamma_s$ slopes. These estimates and parameters have been tested in a clinical dataset of 78 patients in the past, with excellent fit, demonstrating that the assumptions hold in clinical sputum samples[24]. The estimates are derived as medians of the MCMC posterior distribution, the uncertainty was given by 95% credible intervals (CrIs) calculated from the 2.5% and 97.5% quantiles of the MCMC parameter posterior distribution. MCMC convergence was assessed visually and by using the chain convergence diagnostic tools in the R coda package.

**Identification of biomarkers predicting outcomes in derivation dataset**. Identification of biomarkers that best predicted therapy outcomes was carried out using classification and regression criteria (CART) of Breiman et al.[47]. Using the derivation dataset, we examined all demographic, clinical, and radiological factors, as well as model-derived $\gamma_s$ and $\gamma_f$ slopes and the initial bacterial burdens ($B(0)$), as potential predictors of outcome. Outcome was defined as either therapy success at end of therapy, or therapy failure (failure at the end of treatment or relapse), or relapse. The steps we followed were implemented by two independent investigators in R (Rpart) and Salford software, and have been described in detail in the past[39].

First, CART analysis was used to identify and rank the top predictors of therapy failure and relapse. Second, we used clustering to characterize the relationship of the top predictors for each specific treatment outcome, and also identified the statistical association[48]. TTP trajectories were clustered using the K-means algorithm implemented in the KML-package in R[48]. The 6-month TTP data for each cluster was

reduced to derive (i) the 4-month slopes [using the first 4 months accrued data] and (ii) then 2-month slopes (based on the first eight-week accrued data). The model was fitted to data for each separate cluster and their respective reduced subsets.

Finally, we utilized Markov chain Monte Carlo simulations of time-to-extinction in tandem with CART to identify slope thresholds and initial bacterial burden that best classified relapses and therapy failure[46].

**Mathematical simulations for indeterminate data zones**. We computed 10,000 bacteria trajectories to simulate different treatment outcomes. The initial bacterial burdens based on the range in derivation data set of between 3–7 $\log_{10}$ CFU/mL and $\gamma$-slopes between 0.05 to 0.5 $\log_{10}$ CFUs/day, were varied simultaneously, with the rest of all model parameters held constant. TTE for each separate trajectory was computed. The TTE values define the transcritical bifurcation points that explains when the Mtb NRP stable state switches to extinction. Regions of time within which bacteria subpopulations would go extinct were constructed and partitioned to reflect the expected clinical treatment duration intervals.

**Sensitivity analysis for treatment duration**. Monte-Carlo experiments were carried out to identify changes in $\gamma_s$ values that resulted in treatment duration shortening (2 and 4 months) and those that led to prolonged treatment duration (7, 8 and 9 months). Magnitudes that correspond to these treatment end-points were determined relative to different categories of patient initial bacterial load, (i) high (>5·0 $\log_{10}$ CFU/mL), (ii) medium (3·5–5·0 $\log_{10}$ CFU/mL) and (iii) low (<3·5 $\log_{10}$ CFU/mL). These bounds were selected to toggle between CART discrete bounds and sweep across continuous patient CFU burdens to examine effect of different slope magnitudes on outcome for the defined therapy durations.

**Validation of identified biomarkers**. Individual patient TTP trajectories were fitted to the model to identify the corresponding $\gamma_s$ and $\gamma_f$ in the validation datasets. The accuracy, sensitivity, and specificity of biomarkers derived in the derivation dataset were calculated using the validation dataset for cure, relapse, or therapy failure, for 6 and 4-months duration of therapy. The definitions for cure, relapse, and therapy failure used were those defined by the REMoxTB clinical trial protocol[3]. We used the standard statistical and clinical definitions for sensitivity, specificity, accuracy, and the number needed to diagnose failure and relapse[49,50].

**Statistics and reproducibility**. Mean values between groups were compared using Student's t-test or analysis of variance (ANOVA) F-test, while the Mann-Whitney test was used for proportions and compare medians from distributions of the fast and slow slopes derived at 2-months, 4-months and 6-months accrued TTP data. Spearman's correlations were used to examine correlation while un-weighted Cohen' Kappa coefficients examined agreements of clinical outcomes derived from REMoxTB study definition versus those derived from the model based on time-to-extinction. All analyses were performed with packages in R.

**Reporting summary**. Further information on research design is available in the Nature Research Reporting Summary linked to this article.

## Data availability
Data underlying the figures in this study are available at the Dryad repository [https://datadryad.org/stash/share/1bUdMU9bsYcJeAzlMtqhUvFIYrboRS9DwH3AgvLYYAk]. Other source data are available from the corresponding author upon reasonable request.

## Code availability
Data were analyzed used R and the data analysis scripts are available from the corresponding author upon reasonable request.

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

## Acknowledgements

We thank Dr. Stephen H Gillepsie and Dr. Patrick Phillips for help with insightful comments on the manuscript, and general advice about the REMoxTB study design and data.

## Author contributions

G.M. performed the mathematical modeling, interpreted the data and wrote the first draft of the manuscript; J.G.P. performed sample size calculations, CART and statistical analyses, and designed the sample size calculator; T.G. oversaw the conduct of the study, led the data interpretation, application of criteria for biomarkers, and the clinical meaning. All three authors wrote the manuscript, revised it, and all approved the final submitted version.

## Competing interests

The authors declare no competing interests.
