## [Transparent Peer Review File · Communications Biology]

Reviewers' comments:

Reviewer #1 (Remarks to the Author):

The authors present a dynamic modeling and decision tree analysis of phase III REMoxTB data based on a bacterial kill curve slope-based biomarker for TB drug treatment outcomes. The results have potential general application to shortening the clinical development time of novel TB regimens, and to individualized therapy.

The manuscript can be improved with a more detailed description of the model parameter estimation and results, and better support for the biological interpretation of the slopes.

Comments

line 59: is it "... phase I clinical trials" or phase II - as EBA studies are typically the latter?

lines 88-89: The authors note time-to-extinction as the required minimum duration of therapy. Some additional explanation regarding the role of the host immune response would better support this idea. The authors seem to imply that relapse or treatment failure is independent of possible host effects, such as in latent TB. Could it be that non-zero bacterial load may be necessary but perhaps not a sufficient condition for relapse?

line 93: What were the criteria used to determine which data were insufficient and removed? In the methods, there is a mention of more than four data points - but also of criteria based on contamination (line 412). Also, does the timing matter, i.e. early or late data? There are two slopes and a subpopulation fraction to consider, and on what basis are the number of data points deemed sufficient to identify these parameters? Does this number include the initial conditions or baseline measurements?

line 96: The TTP measurements were converted to CFU's. Were there not CFU measurements also collected in the REMoxTB studies that could be used directly, or to characterize uncertainty in the TTP to CFU conversion? I understand the comparison made using a separate study to verify time-independence of the conversion formula - which is a fairly important result itself - and deserves some discussion e.g., could it be drug-dependent? But why not use CFU's from the REMoxTB results? If they were not available, then that point could be made explicit.

lines 100-102: The kill-rate slopes are interpreted as corresponding to fast-replicating and semi-dormant or non-replicating M.tb. As the bacteria are from sputum samples, such an interpretation should be better supported by a description of phenotypic variation that may already be established in such sputum samples - particularly the proportions of each. Is there a distinction between the phenotypic states resulting from drug treatment or a particular pathophysiological environment in the lung? In several EBA studies, a biphasic relationship for standard regimen controls can be seen in the first few days. In the results here - the inflection point appears much later - looking more like 2 weeks in Figures S2. How would the timing of such phases correspond to these different subpopulations in sputum?

Additionally, in regard to the model equations, how were the initial values for separate fast and slow subpopulations determined (it appears to be as an estimated fraction f - but I'm not exactly clear on that), and for the growth rates r_f , r_s , and K_{max} ? I don't see the latter values listed in the manuscript. It appears that these were set as point estimates without any accounting for variability. I would think they may be dependent on the patient population, or possibly also on the drug regimen - especially if the r_s were somehow dependent on the particular drugs or treatment. It does not appear that these parameters were included for estimation - some discussion of this would help support leaving them out of the estimation process. A Table showing model parameter values - what was estimated and what was fixed would be helpful to understanding the results.

CART: Mention is made of implementation by two independent investigators with different software. Were the results the same? line 147: It is noted that all clinical and laboratory features were included, but I see only model parameters discussed. For example, where did specific other features such as MIC's, HIV status, cavitation, male and female rank?

lines 141: Methods on the MCMC sampling. It would be helpful to better describe the prior and posterior distributions. For example, what about upper and lower bounds on the uniform priors - how were they chosen and how did such choices affect convergence? Were the posterior distributions multi-modal? What were the Gelman-Rubin diagnostic values?

line 435: please describe each term in the equation, with the appropriate units to the values given.

line 518: the TTE values are described as defining bifurcation points. Some additional explanation would help here. It seems though as they are just the time points at which the CFU trajectories pass the 10^{-2} CFU threshold value. For example - wouldn't a small increase or decrease in the value of a slope parameter just lead to a smaller or larger TTE value rather than a qualitative change in outcome?

Discussion:

As there was an extensive statistical analysis that included the REMox data by Imperial et al, some comparison of these biomarker results with the major conclusions from that paper (for example, differences between male and female outcomes, percentage of patients for which the four-month treatment was effective) would add additional context and support for the slopes as a treatment biomarker.

The results depend on (and support) sputum samples as an adequate and representative biological sample for TB drug testing. Some discussion of the extent to which sputum samples are considered appropriate in general, including for example the quality and volume as treatment progresses over time, would help to clarify the limitations to these results.

The fractions of semi-dormant bacteria in Table S1 seem very high compared to other estimates (say of persister subpopulations of 0.1% or less). Some discussion on how the interpretation of the model slopes compares to other subpopulation descriptions would be helpful.

Some discussion of optimal experimental design for this model could be useful for future clinical trials that could include data support for minimizing uncertainty in the slopes.

Reviewer #2 (Remarks to the Author):

This manuscript describes an in silico effort to establish an improved method to predict treatment outcome in human tuberculosis (TB) patients based on sputum bacterial loads, which takes advantage of fitting longitudinally sampled sputum bacterial load data to differential equations that model two co-existing Mycobacterium tuberculosis (Mtb) subpopulations. The metric developed by the authors

(based on “gamma-s”, an estimated kill rate of the slow/non-replicating Mtb subpopulation) appears to be significantly more accurate at distinguishing successful cure from treatment failure or relapse than the conventional early bactericidal activity (EBA) or two-month sputum conversion rate metrics. This metric has the potential to be transformative for the field of TB therapy if it reproduces in other future studies. However, the manuscript in its current form is missing critical explanations and statistical assessments in several key areas, which limits the ability of readers to adequately assess the validity of model and reproduce it for future studies.

Major concerns:

- The explanation and justification for the calculation of CFU/mL based on the “time to positivity” (TPP) values measured from sputum incomplete and confusing. In line 131-132, the authors state that the “TPP versus CFU/mL relationship could change with time during treatment.” If this is in fact the case, then how were the authors able to establish that the Gompertz model and the equation parameters used to convert between TPP and CFU/mL would be optimal for the entire duration of therapy? Statistical assessments evaluating the goodness of fit for this model in patient data over a timecourse of treatment appear to be missing. Visual inspection of the graphs presented comparing TPP to CFU/mL in Figure S1 (at one time point? It’s unclear from the figure caption) suggests substantial heterogeneity in the TPP vs CFU/mL relationship, especially for slow/non-replicating bacteria (panels C & D) and for “combined subpopulations” (panels E & F). Furthermore, visual inspection of panels G and H makes it difficult to see any association between the measured TPPs and measured CFU/mL values.
 - o A more convincing quantification of the validity of the CFU/mL calculation formula would be an assessment of the degree of concordance between calculated CFU/mL values derived from TPP input data and the corresponding experimentally measured CFU/mL values from the same patient measured at the same timepoint, for multiple timepoints of treatment.
- There is insufficient clarity on how key metrics were calculated from the bacterial load input data, and what assumptions were imposed to enable this calculation:
 - o The model equations used to estimate the kill rates for the Mtb subpopulations (“gamma-s” and “gamma-f” in equations 2 and 3 in line 468) include 3 other unknown parameters, and the description for how these other parameters were estimated from the data is incomplete and difficult to follow. In the absence of additional descriptions for how these parameters were defined, it seems as if the model were under-determined. There also appears to be a discrepancy between the total of 5 parameters featured in equations 2 and 3 and the parameters estimated in Table S1, which only features 4 parameters, one of which is not described in equations 2 or 3. This makes it difficult to understand how several of the parameters from equations 2 and 3 were estimated.
 - * How are the authors able to estimate how big a contribution slow/non-replicating subpopulation makes relative to the whole bacterial burden? Were some of the parameter values estimated from accrued data and then used to help constrain the estimation of the other parameters from individual patient data?
 - * How was the variable “f” in Table S1 estimated from the input data? Was this variable assumed to be consistent across all patients? Irrespective of strain type infected?
 - * How much input data for each individual is required to generate an adequate estimate of “gamma-s”? How many timepoints?
 - * How is the goodness of fit for the model fitting/parameter estimation gauged?
- The statistical tests and associated explanations used in the “Time-to-extinction versus clinical trial-based outcome definitions” section (starting line 129) are wholly appropriate for the assessment intended.
 - o For example, the Cohen’s kappa metric is not designed to assess accuracy relative to a ‘gold standard.’ Sensitivity and specificity would be more appropriate here, both because of their broader use and because of their direct assessment true positive and true negative rates.
 - * In addition, it is not spelled out what the “standard clinical definitions” (line 138) the authors are comparing against, which makes this assessment harder to evaluate.
- There is an inadequate assessment for the claim made in lines 170-171 that the in silico model explained the experimentally observed patient data “well.” No statistics are provided, and visual

inspection of Figure S2 shows a great deal of heterogeneity in the patient data, which incompletely overlaps the 95% confidence intervals of the model fits.

o It is also unclear the extent to which the 4 clusters identified from the TPP trajectories align with the clinical definitions of cure/slow cure/relapse/treatment failure based on the clinical definitions. No quantitative assessment is made.

Minor concerns:

- The text in Figure 4H and 4I are very small and difficult to read.
- The figure reference in line 189 should be Figure 4H, not Figure 4I, since it is referring to “gamma-s”.
- What is the “number needed to diagnose” variable on line 256? This parameter is incompletely described.
- The equation used to convert TTP to CFU/mL on line 435 should have all variables clearly defined. In its current form, it’s unclear what the variables “F” or “x” represent.
- The values in Table S2 are not labeled with units, making it unclear what the values are describing.
- The figure labels in Figure S1 are small and hard to read.
- The figure caption for Figure S3 is incorrectly labeled.

RE: COMMSBIO-20-3180-T "Bacterial load slopes as biomarkers of tuberculosis therapy success, failure, and relapse"

We would like to thank you and the reviewers for the detailed comments. We have revised the manuscript taking into account the points you and reviewers raised, and also provide a point-by-point response below where we have re-numbered the critiques and responses for ease of communication:

Reviewer #1 (Remarks to the Author):

Query #1: The authors present a dynamic modeling and decision tree analysis of phase III REMoxTB data based on a bacterial kill curve slope-based biomarker for TB drug treatment outcomes. The results have potential general application to shortening the clinical development time of novel TB regimens, and to individualized therapy.

The manuscript can be improved with a more detailed description of the model parameter estimation and results, and better support for the biological interpretation of the slopes.

Response #1: We would like to thank the reviewer for the general comments, and for the comments to improve the descriptions of model parameter estimations and results. We have added more details on model parameter description and on the use of slopes

Comments

Query #2: line 59: is it "... phase I clinical trials" or phase II - as EBA studies are typically the latter?

Response #2: We have corrected our terminology; 14-day EBA is indeed a phase IIa while the 8-week study is a phase IIb, as reflected on EBA and 8 week studies on clinicaltrials.gov

Query #3: lines 88-89: The authors note time-to-extinction as the required minimum duration of therapy. Some additional explanation regarding the role of the host immune response would better support this idea. The authors seem to imply that relapse or treatment failure is independent of possible host effects, such as in latent TB. Could it be that non-zero bacterial load may be necessary but perhaps not a sufficient condition for relapse?

Response #3: We agree. We added the sentence "However, some patients who do not reach bacterial-population extinction as a result of TB therapy but can still achieve relapse-free cure because of immune-containment, and in that fraction of patients host responses can mop up remaining bacteria to achieve cure or prevent relapse."

Query #4: line 93: What were the criteria used to determine which data were insufficient and removed? In the methods, there is a mention of more than four data points - but also of criteria based on contamination (line 412). Also, does the timing matter, i.e. early or late data? There are two slopes and a subpopulation fraction to consider, and on what basis are the number of data points deemed sufficient to identify these parameters? Does this number include the initial conditions or baseline measurements?

Response #4: We have changed the sentences describing inclusion and exclusion criteria to improve clarity. The sentence in lines 92-94 now reads: "(i) majority of sputum samples that were contaminated [TTP<4days], or missing, or (ii) ≤ 4 data-points within the first 8 weeks (i.e., data-points fewer than ODE model parameters minus one) excluding the baseline value were removed." **Note:** The baseline bacteria TTP value is assumed as the initial B[0] value. Also, data points within the first 4 weeks (28 days of treatment) were adequate to estimate the two slopes, since the second slope becomes dominant between 7-14 days of treatment.

Query #5: line 96: The TTP measurements were converted to CFU's. Were there not CFU measurements also collected in the REMoxTB studies that could be used directly, or to characterize uncertainty in the TTP to CFU conversion? I understand the comparison made using a separate study to verify time-independence of the conversion formula -which is a fairly important result itself - and deserves some discussion e.g., could it be drug-dependent? But why not use CFU's from the REMoxTB results? If they were not available, then that point could be made explicit.

Response #5: Unfortunately, there were no accompanying CFUs in the REMoxTB study; indeed these would have been the best to characterize uncertainty in TTP to CFU conversion. We discuss this in the limitations paragraph where we have added the lines 101-103 and lines 297-307 in the discussion

REMoxTB TTP-CFU data-pairs would have been the best at characterizing the uncertainty in TTP conversion to CFU conversion. However, we tested the time-to-extinction based definitions of cure derived using this TTP-CFU conversion from the prior study to those observed using the REMoxTB clinical trial protocol definitions and patient microbiology and identified a Spearman rank correlation of 1.0 [$p=0.017$]. Thus our TTP-CFU conversion and slopes derived from it were robust."

Query #6: lines 100-102: The kill-rate slopes are interpreted as corresponding to fast-replicating and semi-dormant or non-replicating Mtb. As the bacteria are from sputum samples, such an interpretation should be better supported by a description of phenotypic variation that may already be established in such sputum samples - particularly the proportions of each. Is there a distinction between the phenotypic states resulting from drug treatment or a particular pathophysiological environment in the lung? In several EBA studies, a biphasic relationship for standard regimen controls can be seen in the first few days. In the results here - the inflection point appears much later - looking more like 2 weeks in Figures S2. How would the timing of such phases correspond to these different subpopulations in sputum?

Response #6:

This is an excellent question, and foundational to our approach. In response to this, as well as queries #14, YY and ZZ below, our opening paragraph in the discussion now reads as follows in lines 297-307:

“First, our model estimates of initial proportion of non-replicating persistent/semidormant of 1-25% of fast replicating [Table S1], and the microbial kill slope inflection point of about 14 days would seem to be different from the proportions of ~0.1% NRP [slow] of total bacilli and early bactericidal effect studies in which the inflection point is 3 days [7-9, 26-28]. However, in studies that have directly visualized sputa of patients using lipid bodies [LB] to identify slow bacilli [and thus nicknamed “fat and lazy” by Garton et al] and confirmed by bacterial *tgs1* gene expression, “the frequency of LB-positive cells varied from 3% to 86%” [29]. Estimates of LBs by Sloan et al were 28% (interquartile range: 13–54)% [44]. Thus, our model estimates of proportion of slow bacilli are in the same range as LBs visualized in sputum. The Sloan study also followed the change in LBs on standard therapy and identified the inflection point in proportion LBs was after 14-21 days, consistent with our inflection points slow bacilli in **Figure S2**. Thus, our model parameter estimates were highly consistent with the biology of bacillary populations, and their changes with therapy. This is the major strength of our approach, which is that they are mechanistic and based on the biology observed in patients.”

Query #7: Additionally, in regard to the model equations, how were the initial values for separate fast and slow subpopulations determined (it appears to be as an estimated fraction f - but I'm not exactly clear on that), and for the growth rates r_f , r_s , and K_{max} ? I don't see the latter values listed in the manuscript. It appears that these were set as point estimates without any accounting for variability. I would think they may be dependent on the patient population, or possibly also on the drug regimen - especially if the r_s were somehow dependent on the particular drugs or treatment.

It does not appear that these parameters were included for estimation - some discussion of this would help support leaving them out of the estimation process. A Table showing model parameter values - what was estimated and what was fixed would be helpful to understanding the results.

Response #7:

We agree with the review this information was left out and it is our mistake. We have now provided details how these were estimated and a reference to our previous study.

The initial of the fast population is first estimated as the mean value baseline CFU values at $t=0$, then the slow population is assumed to be $B_s(0)=f*B(0)$, and f is estimated during modeling fitting, our assumption is that initially there is small population of slow replicating bacteria that

will eventually become apparent as the therapy depletes the Bf population. This phenomenon is the one driving the change from fast declining slope to slow declining slope.

The r_f , r_s and K_{max} parameters were derived using our hollow-fiber experiments, in which clinical isolates were grown under conditions to reproduce log-phase growth and slow non-replicating growth. From these experiments we also derived K_{max} . Details for these are published in Magombedze et al CID 2018. This information is now included in the manuscript, Table S6

In model fitting these were held to the median values, however, in monte-carlo simulations, they were varied between their lower and upper bounds credible intervals, hence the wide confidence bounds shown in Fig S2.

Query #8: CART: Mention is made of implementation by two independent investigators with different software. Were the results the same? line 147: It is noted that all clinical and laboratory features were included, but I see only model parameters discussed. For example, where did specific other features such as MIC's, HIV status, cavitation, male and female rank?

Response #8: We did not have access to any MIC or drug concentration values from these datasets. However, HIV status, cavitation, and anthropometric features were available and included in the CART. In the results section, "Predictors of outcome in derivation dataset" section, in lines 152-154, we have added the sentence:

"Similarly, features such as HIV status, cavitation, biological sex [male or female] ranked low and had variable importance score below 10%."

Query #9: lines 141: Methods on the MCMC sampling. It would be helpful to better describe the prior and posterior distributions. For example, what about upper and lower bounds on the uniform priors -how were they chosen and how did such choices affect convergence? Were the posterior distributions multi-modal? What were the Gelman-Rubin diagnostic values?

Response #9:

We have now added the parameter prior ranges, Table S1 and Table S6. The parameters chains were visually examined for convergence and we also used the R Coda package with Gelman-Rubin diagnostics (<1.1) for chain convergence. These details were not included in the MS because we considered this a standard exercise worthy not reporting. The posterior distributions were not multi-modal. Actually, the parameters posteriors are the ones shown in Figs 4A-G.

Query #10: line 435: please describe each term in the equation, with the appropriate units to the values given.

Response #10:

$$F = \alpha e^{-\beta x + \gamma}$$

F is CFU/ml, X is TTP (number of days the MGIT indicate presence of viable bacteria). The formulae is to the corresponding CFU/mL based on time taken for MGIT to indicate growth. If TTP is very small (short time to positivity) it means many viable bacteria present, if it is large value (42 days) indicates very minute viable bacteria that has to take time to indicate growth. The parameter α -represent the corresponding CFU quantity at $x=0$ [units CFU/mL], of when TTP take less than a day to turn positive, while β is a rate value for apportioning changing TTP to the corresponding CFU value [TTP conversion rate per day], and γ is constant to adjust for conversion and capture wide data variability.

For example consider $\frac{dF}{dX} = \beta F$, integrating gives $F = F(0)e^{\beta X}$, where $\alpha = F(0)$, which is the simple model and then adding a constant in the exponent gives $F = \alpha e^{\beta X + \gamma}$, note here γ essentially adds more variability, but based on AIC score both models fit the data well and have similar parameters. Using either of the model will not change the conversion. And in our conversion we used estimates with combined log-phase and non-replicating bacteria

Model parameters are now included in the MS (see Table S5). Also we included an alternative model with $\gamma=0$, and compared these two models. Note here we provided AIC scores for model comparison and goodness of fit.

Query #11: line 518: the TTE values are described as defining bifurcation points. Some additional explanation would help here. It seems though as they are just the time points at which the CFU trajectories pass the 10^{-2} CFU threshold value. For example - wouldn't a small increase or decrease in the value of a slope parameter just lead to a smaller or larger TTE value rather than a qualitative change in outcome?

Response #11:

The explanation by the review is correct. The bifurcation can be explained by the changing values in the slopes. But at that point both bacteria populations go to zero. However, in the most sensitive assay, the MGIT readout, this corresponds to 0.6 CFU/ml, experimentally <1 CFU is zero CFUs, we therefore instead in our simulations used a more stringent cut-off of 0.01. Here in our responses we have included a Fig to illustrate this point. Fig A, shows both Bf & Bs going to zero (extinction), in Fig B, we show that γ_s for Bs drives persistence and we see the γ_s value that defines the bifurcation from persistence to extinction. Now in C, we show the relationship between γ_s the bifurcation parameter with time to extinction

Query #12: Discussion:

As there was an extensive statistical analysis that included the REMox data by Imperial et al, some comparison of these biomarker results with the major conclusions from that paper (for example, differences between male and female outcomes, percentage of patients for which the four-month treatment was effective) would add additional context and support for the slopes as a treatment biomarker.

Response #12: We agree. We have now added the following sentences to give context, in the discussion section, lines 357-361

“As regards to clinical practice, our findings add to the recent discovery that initial Mtb burden can be used to determine patients who can benefit from 4-month duration therapy [47]. Imperial et al used standard inferential statistics found that non-adherence was the most significant factor leading to poor outcomes [hazard ratio 5.9], and that low initial bacterial and disease burden were the most important determinants of optimum duration of therapy [47].”

Query #13: The results depend on (and support) sputum samples as an adequate and representative biological sample for TB drug testing. Some discussion of the extent to which sputum samples are considered appropriate in general, including for example the quality and volume as treatment progresses over time, would help to clarify the limitations to these results.

Response #13: The reviewer is correct. However, we depended on the primary trial’s description and samples passing QC. We have added this as a limitation in lines 393-397: “Fourth, we relied on data supplied by the clinical trial team, and thus could not assess the quality and volume of sputum samples as treatment progressed, and several factors associated sputum collection effect of bacterial burden. This limitation however is somewhat mitigated by the finding that model fits did not change from start of therapy to end of therapy.”

Query #14: The fractions of semi-dormant bacteria in Table S1 seem very high compared to other estimates (say of persister subpopulations of 0.1% or less). Some discussion on how the interpretation of the model slopes compares to other subpopulation descriptions would be helpful.

Response #14: See response #6.

Query #15: Some discussion of optimal experimental design for this model could be useful for future clinical trials that could include data support for minimizing uncertainty in the slopes.

Response #15: This is an excellent point raised. We have now added lines 350-353 in discussion “Moreover, these data and slopes can be used for optimal design by identifying optimal sampling times of sputa [TTP], and optimal number of samples that minimize uncertainty in the slopes in future clinical trials.”

Reviewer #2 (Remarks to the Author):

Query #16: This manuscript describes an in silico effort to establish an improved method to predict treatment outcome in human tuberculosis (TB) patients based on sputum bacterial loads, which takes advantage of fitting longitudinally sampled sputum bacterial load data to differential equations that model two co-existing Mycobacterium tuberculosis (Mtb) subpopulations. The metric developed by the authors (based on “gamma-s”, an estimated kill rate of the slow/non-replicating Mtb subpopulation) appears to be significantly more accurate at distinguishing successful cure from treatment failure or relapse than the conventional early bactericidal activity (EBA) or two-month sputum conversion rate metrics. This metric has the potential to be transformative for the field of TB therapy if it reproduces in other future studies. However, the manuscript in its current form is missing critical explanations and statistical assessments in several key areas, which limits the ability of readers to adequately assess the validity of model and reproduce it for future studies.

Response #16: We would like to thank the reviewer.

Query #17: Major concerns:

- The explanation and justification for the calculation of CFU/mL based on the “time to positivity” (TTP) values measured from sputum incomplete and confusing. In line 131-132, the authors state that the “TPP versus CFU/mL relationship could change with time during treatment.” If this is in fact the case, then how were the authors able to establish that the Gompertz model and the equation parameters used to convert between TPP and CFU/mL would be optimal for the entire duration of therapy?

Response

To derive the TTP to CFU conversion model, we used hollow fiber data with more than 600 data pairs for both bacterial subpopulations-that is log phase and non-replicating phases (See response to Query #10). In the TB in-vitro hollow fiber experiment, these different bacterial populations were cultured for prolonged periods of time up to 56 days, using similar methods

used in TB clinical studies though in clinical studies this is done up to 42 days. Both the CFU culture and MGIT methods were applied for data coming from the same experiment. Now, the aspect of time is very simple, time to positive correlates with time spend on therapy therefore culturable bacteria still viable. In the clinical trial TPPs were measured during therapy at different time points, and this approach will give the equivalent CFU values at each point.

Query: Statistical assessments evaluating the goodness of fit for this model in patient data over a time course of treatment appear to be missing. Visual inspection of the graphs presented comparing TPP to CFU/mL in Figure S1 (at one time point? It's unclear from the figure caption) suggests substantial heterogeneity in the TPP vs CFU/mL relationship, especially for slow/non-replicating bacteria (panels C & D) and for "combined subpopulations" (panels E & F). Furthermore, visual inspection of panels G and H makes it difficult to see any association between the measured TPPs and measured CFU/mL values.

Response: We have added the goodness of fit measures in the Supp File, See Table S5 Note: In Fig S1, two models were used (i) $F = \alpha \exp(\beta * x)$ and (ii) $F = \alpha \exp(\beta * x + \gamma)$. And the model with gamma explains more heterogeneity (as shown by the 95% CIs) Figs D & H.

For Figs S1 G and H, we apologize, we should have done a better job to explain these. (i) The TPPs in Fig S1G are actual clinical measurements (N=70 patients) from an independent study and (ii) in H we show the CFU we obtained using the conversion model and these are consistent with what we expected. Also importantly the same conversion formula can be used to convert CFU back to TPP. It is important to realize that when you see a large TPP that should correspond to a small CFU value. Since it will take a culture with less bacteria to indicate growth in a MGIT machine. Figs G and H are what we expected.

Query: A more convincing quantification of the validity of the CFU/mL calculation formula would be an assessment of the degree of concordance between calculated CFU/mL values derived from TPP input data and the corresponding experimentally measured CFU/mL values from the same patient measured at the same timepoint, for multiple timepoints of treatment.

Response: We agree, unfortunately we did not have such data since in clinical trials TTP measurements are preferred. However, this is what we actually sort to achieve using Hollow-fiber TB data, which does a good job to imitate human TB treatment. Then Fig S1 **G** (TTP clinical data) and **H** (corresponding CFUs obtained by model conversion), gives a picture that is consistent with actual TB treatment readouts and treatment outcomes.

Query #18: • There is insufficient clarity on how key metrics were calculated from the bacterial load input data, and what assumptions were imposed to enable this calculation:

o The model equations used to estimate the kill rates for the Mtb subpopulations ("gamma-s" and "gamma-f" in equations 2 and 3 in line 468) include 3 other unknown parameters, and the

description for how these other parameters were estimated from the data is incomplete and difficult to follow. In the absence of additional descriptions for how these parameters were defined, it seems as if the model were under-determined.

Response

We have now added information in the MS with details on how parameters were estimated. In summary, two different data sets were used (details published in Magombedze et al., 2018 CID). Data fitting was done in two steps (ii) in step-1 with Hollow-fiber data with the bacteria growth to imitate log phase growth and the non-replicating phase. These data were used to estimate the rates r_f and r_s , and the K_{max} . Note the Hollow fiber experiments to estimate bacteria growth were done with drug susceptible clinical isolates to estimate bacterial replication in actual patients (ii) in step-2, the rates from step-1 were fixed at obtained values, and parameters f , g_f and g_s were estimate using the clinical data.

Q: There also appears to be a discrepancy between the total of 5 parameters featured in equations 2 and 3 and the parameters estimated in Table S1, which only features 4 parameters, one of which is not described in equations 2 or 3. This makes it difficult to understand how several of the parameters from equations 2 and 3 were estimated.

Response

The parameters r_f and r_s , and the K_{max} were estimated in our study Magombedze 2018 CID, we have now added this clarification. In the table we only included parameters (f , g_f and g_s) that were estimated using the clinical data.

Q* How are the authors able to estimate how big a contribution slow/non-replicating subpopulation makes relative to the whole bacterial burden? Were some of the parameter values estimated from accrued data and then used to help constrain the estimation of the other parameters from individual patient data?

Response:

This is a very good question. Our assumption is that before treatment, the initial bacteria burden $B(0)$ is a sum of $B_f(0)$ and $B_s(0)$, that is $B(0)=B_f(0)+B_s(0)$, however, the population is predominantly $B_f(0)$, therefore $B_f(0)$ is assumed approximately equal to $B(0)$. Then $B_s(0)$ is assume to be $f*B_f(0)$. The f is estimated during fitting to estimate the initial $B_s(0)$ that is unknown. See the Fig below, the unshaded circle is $B_f(0)$, but notice a line from the third circle (in purple) extrapolates back in time to that point (about 4 log₁₀ CFUs), indicating persistence demonstrated by the slow slope is driven by this population that get unmasked in the course of treatment as B_f population gets rapidly killed by the drugs.

* How was the variable “ f ” in Table S1 estimated from the input data? Was this variable assumed to be consistent across all patients? Irrespective of strain type infected?

Response: Please see the previous responses

* How much input data for each individual is required to generate an adequate estimate of “ γ - s ”? How many timepoints?

Response

For γ - s at 2M we used 8 data points (weekly readouts for 2 months). However, we estimated 3 parameters, therefore 4 data points without the baseline data $t=0$ (5 data points if including the baseline data point) is required to estimate these 3 parameters. The baseline values is used as $B(0)$.

* How is the goodness of fit for the model fitting/parameter estimation gauged?

Response: We used MCMC with Bayesian statistics assuming a gaussian likelihood to estimate model parameters. Likelihoods and Akaike Information criteria are quantitative measures of model of fit. MCMC chain converge within 20-40 acceptance rate demonstrates good model fitting. That is basically the approach we took. In addition, we provided the 95% credible intervals of the estimates to demonstrate uncertainty in the parameter estimates.

- The statistical tests and associated explanations used in the “Time-to-extinction versus clinical trial-based outcome definitions” section (starting line 129) are wholly appropriate for the assessment intended.

- o For example, the Cohen’s kappa metric is not designed to assess accuracy relative to a ‘gold standard.’ Sensitivity and specificity would be more appropriate here, both because of their broader use and because of their direct assessment true positive and true negative rates.

Response: We agree with reviewer. However, using these time-to-extinction metrics in the validation dataset, whereby we used standard definitions of clinical outcome [ie., the gold standard] and we demonstrate robust sensitivities and specificities.

Q* In addition, it is not spelled out what the “standard clinical definitions” (line 138) the authors are comparing against, which makes this assessment harder to evaluate.

Response: The standard clinical definitions of cure at end of therapy and relapse, and sterilizing activity are given in the introduction section for the manuscript.

Q• There is an inadequate assessment for the claim made in lines 170-171 that the in silico model explained the experimentally observed patient data “well.” No statistics are provided, and visual inspection of Figure S2 shows a great deal of heterogeneity in the patient data, which incompletely overlaps the 95% confidence intervals of the model fits.

Response:

We used MCMC with Bayesian statistics with a gaussian likelihood for modeling fitting. As standard exercise we used visual inspection to assess MCMC chain convergence and R code diagnostic (Gilmen Rubin diagnostics with a scale reduction factor <1.1 comparing two MCMC chains) tools (the parameter posteriors are the distributions given in Fig 2). This is not reported because we believe its standard practice to assess MCMC chain convergence.

Q It is also unclear the extent to which the 4 clusters identified from the TPP trajectories align with the clinical definitions of cure/slow cure/relapse/treatment failure based on the clinical definitions. No quantitative assessment is made.

Response

In the leadup to the clustering results we added the following sentences “Cluster analysis is an agnostic, quantitative and unsupervised machine-learning task. In this case, TPP trajectories and time-to-extinction in the derivation dataset were grouped into four distinct homogenous groups based on a K-means algorithm in the 238 patients, as shown in **Figure 3.**”

The clusters matched exactly the clinical groupings for cure, relapse and failure. As explained in Fig 1. The cure (cluster 1) and slow cure (cluster 2) clusters are the clinical cure however the clusters show that they are individuals that get cured relatively faster. And the cluster 3 and cluster 4 correspond with the clinical relapse and failure clusters.

Minor concerns:

- The text in Figure 4H and 4I are very small and difficult to read.

Response: This has been corrected

- The figure reference in line 189 should be Figure 4H, not Figure 4I, since it is referring to “gamma-s”.

Response: This has been corrected

- What is the “number needed to diagnose” variable on line 256? This parameter is incompletely described.

Response: The NND is number needed to diagnose and which is given by $1/(\text{Youden Index})$, and Youden index = (Sens+Spec-1). The NND is a standard clinical variable that was used here.

- The equation used to convert TTP to CFU/mL on line 435 should have all variables clearly defined. In its current form, it’s unclear what the variables “F” or “x” represent.

Response : This is now included

- The values in Table S2 are not labeled with units, making it unclear what the values are describing.

Response: Units are now included

- The figure labels in Figure S1 are small and hard to read.

Response: Labels font was increased

- The figure caption for Figure S3 is incorrectly labeled.

Response: This is now corrected

REVIEWERS' COMMENTS:

Reviewer #1 (Remarks to the Author):

The authors have addressed all of my comments, with the additional details providing a more clear picture of the method and results.

Reviewer #2 (Remarks to the Author):

The revisions that the authors have made have significantly improved the clarity of the study. I have a few remaining minor points to note intended to further facilitate clarity:

- Line 103: would be worth it to explicitly state that NRP is the acronym of non-replicating bacteria (this is currently only implied).
- Line 171: the sentence "...patients with less than 2 or more missing observations..." is a bit confusing. Do you mean "less than 2 missing observations" or something else?
- There appears to be some kind of discrepancy between the caption for Figure 4B and what Figure 4B itself is showing. The caption refers to gamma-s, but the axis labels in Figure 4B refer to gamma-f.

RE: COMMSBIO-20-3180-T " Bacterial load slopes as biomarkers of tuberculosis therapy success, failure, and relapse"

We would like to thank the Editor and the reviewers for their comments. We have revised the manuscript considering the suggestions the reviewers raised. Below are our responses to the reviewers comments.

Reviewer #2:

Query #1: Line 103: would be worth it to explicitly state that NRP is the acronym of non-replicating bacteria (this is currently only implied).

Response #1: yes NRP is an acronym for non-replicating persisters/bacteria. This is now included as suggested by the reviewer.

Query #2: Line 171: the sentence "...patients with less than 2 or more missing observations..." is a bit confusing. Do you mean "less than 2 missing observations" or something else?

Response #2: We meant 2 missing observations, and this was corrected accordingly

Query #3: There appears to be some kind of discrepancy between the caption for Figure 4B and what Figure 4B itself is showing. The caption refers to gamma-s, but the axis labels in Figure 4B refer to gamma-f.

Response #3: We would like to thank the reviewer for identifying this mistake. It is now corrected. The caption and the Figure labeling are not consistent.